# Plant-Specific Domains and Fragmented Sequences Imply Non-Canonical Functions in Plant Aminoacyl-tRNA Synthetases

**DOI:** 10.3390/genes11091056

**Published:** 2020-09-07

**Authors:** Yusuke Saga, Moeka Kawashima, Shiho Sakai, Kaori Yamazaki, Misato Kaneko, Moeka Takahashi, Natsuko Sato, Yohei Toyoda, Shohei Takase, Takeshi Nakano, Naoto Kawakami, Tetsuo Kushiro

**Affiliations:** 1School of Agriculture, Meiji University, Kawasaki, Kanagawa 214-8571, Japan; ysaga@meiji.ac.jp (Y.S.); moeka.k22@gmail.com (M.K.); skskaha537@gmail.com (S.S.); kaori.ymzk74@gmail.com (K.Y.); super.mogmaglug@gmail.com (M.K.); m0eka2569@docomo.ne.jp (M.T.); ef171134@meiji.ac.jp (N.S.); meijiyohei812@gmail.com (Y.T.); ef81033@gmail.com (S.T.); kawakami@meiji.ac.jp (N.K.); 2Graduate School of Biostudies, Kyoto University, Kyoto 606-8502, Japan; nakano.takeshi.6x@kyoto-u.ac.jp

**Keywords:** aminoacyl-tRNA synthetase, non-canonical function, histidyl-tRNA synthetase, asparaginyl-tRNA synthetase, plant, species-specific domain

## Abstract

Aminoacyl-tRNA synthetases (aaRSs) play essential roles in protein translation. In addition, numerous aaRSs (mostly in vertebrates) have also been discovered to possess a range of non-canonical functions. Very few studies have been conducted to elucidate or characterize non-canonical functions of plant aaRSs. A genome-wide search for aaRS genes in *Arabidopsis thaliana* revealed a total of 59 aaRS genes. Among them, asparaginyl-tRNA synthetase (AsnRS) was found to possess a WHEP domain inserted into the catalytic domain in a plant-specific manner. This insertion was observed only in the cytosolic isoform. In addition, a long stretch of sequence that exhibited weak homology with histidine ammonia lyase (HAL) was found at the N-terminus of histidyl-tRNA synthetase (HisRS). This HAL-like domain has only been seen in plant HisRS, and only in cytosolic isoforms. Additionally, a number of genes lacking minor or major portions of the full-length aaRS sequence were found. These genes encode 14 aaRS fragments that lack key active site sequences and are likely catalytically null. These identified genes that encode plant-specific additional domains or aaRS fragment sequences are candidates for aaRSs possessing non-canonical functions.

## 1. Introduction

Aminoacyl-tRNA synthetases (aaRSs) play a central role in protein synthesis in all organisms. The activation of amino acids and attachment to tRNAs comprise the canonical functions of aaRSs. Some aaRSs in higher eukaryotes have been found to possess non-canonical functions and to be involved in numerous cellular processes [1]. These non-canonical functions are often exhibited through species-specific appended domains found among aaRSs, which seem to have been added late in evolution. The advantages of such single genes with multiple functions may include the acquisition of higher-order functions with minimal genetic resources. Such species-specific domains possessing potential non-canonical functions include the WHEP domain, a helix-turn-helix domain named after aaRSs (tryptophanyl-tRNA synthetase (TrpRS), HisRS, and glutamyl-prolyl-tRNA synthetase (GluProRS)) that harbor this domain and glutathione *S*-transferase (GST) domains. For example, the WHEP domain in human TrpRS interacts with both DNA-dependent protein kinase (DNA-PKcs) and poly(ADP-ribose) polymerase I (PARP-1) to activate p53 kinase upon emptying its active site [2]. The active site cavities of some aaRSs participate in non-canonical functions by binding to cognate amino acids such as in human leucyl-tRNA synthetase (LeuRS), which can activate the mammalian target of rapamycin (mTOR) pathway [3]. In human TrpRS, the active site binds to Trp residues embedded in cadherin to inhibit cell adhesion thereby antagonizing angiogenesis [4]. The plant metabolite resveratrol has been shown to bind to the active site of tyrosyl-tRNA synthetase (TyrRS), inducing a conformational change that enables binding to PARP-1 to exert its biological activity [5]. Alternative splicing can generate numerous fragmented transcripts, many of which do not contain a full catalytic domain and are catalytically null [6]. These splice variants (SVs) have also been shown to participate in numerous biological processes in humans.

Such non-canonical functions even exist in lower eukaryotes such as fungi [7,8]. Therefore, non-canonical functions may be a common feature of aaRSs that widely exist throughout eukaryota. To date, very little is known about non-canonical functions in plants. Since plants experience a very different life cycle from other eukaryotes, the discovery of novel and unique non-canonical functions is expected. Recently, Arabidopsis aspartyl-tRNA synthetase (AspRS) was shown to participate in disease resistance induced by β-amino butyric acid (BABA) [9]. BABA was demonstrated to bind to the active site of AspRS, triggering the plant defense response against pathogens. Arabidopsis seryl-tRNA synthetase (SerRS) was also found to bind to oxidoreductase BEN1, which is thought to participate in plant hormone brassinosteroid inactivation [10]. The expression of methionyl-tRNA synthetase (MetRS) gene of wheat was upregulated by a mycotoxin deoxynivalenol treatment and conferred resistance to this mycotoxin [11]. A number of characterized aaRS mutants impair particular developmental processes, but do not impair global protein synthesis, which is generally lethal [12,13,14,15,16,17]. These observations suggest that aaRSs in plants may have specific functions other than tRNA aminoacylation.

To shed light on non-canonical functions in plant aaRSs, we surveyed genomic data from the model plant Arabidopsis to search for aaRS genes containing plant-specific domains. During the course of our studies, we noticed that the plant genome contains numerous fragmented aaRS sequences, many of which lack catalytic domains.

## 2. Materials and Methods

### Bioinformatics

*A. thaliana* aaRS genes were retrieved from The Arabidopsis Information Resource (TAIR) database (https://www.arabidopsis.org/) by annotation searches and BLASTp homology searches with previously identified Arabidopsis aaRS sequences as query. Similarly, aaRS genes from rice were obtained from The Rice Annotation Project (RAP-DB) database (https://rapdb.dna.affrc.go.jp/). The aaRS sequences for the other species were downloaded from UniProt database (https://www.uniprot.org/). The obtained sequences were analyzed by InterProScan (http://www.ebi.ac.uk/interpro/) to identify the domain architecture. Pairwise sequence identities were calculated using GENETYX-MAC (ver. 16.0.9, GENETYX Corporation, Tokyo, Japan). Multiple sequence alignments were carried out with MUSCLE (ver. 3.8.31) (https://www.drive5.com/muscle/) using default parameters. Phylogenetic analyses using maximum likelihood method were done with RAxML-NG (ver. 0.9.0) (https://github.com/amkozlov/raxml-ng) with 1000 bootstrap replicates, employing optimal amino acid substitution model for each aaRS alignment. The optimal models were selected using ModelTest-NG (ver. 0.1.6) (https://github.com/ddarriba/modeltest) (Appendix A). Neighbor-Joining (NJ) trees were constructed with 1000 bootstrap replicates using MEGA X on Poisson distances. Phylogenetic trees were drawn with iTOL. Prediction of targeting sequences was carried out with TargetP-2.0. For homology modeling of SYNC3, BLASTp was used to search suitable templates in the PDB. Structures of AsnRS from *Pyrococcus horikoshii* (1X54) and a WHEP domain from human TrpRS (1R6T) were used as templates to build SYNC3 structure. The structure of AspRS bound with tRNA (1EFW) was used as a template for modeling bound tRNA. Homology modeling was performed using MODELLER (ver. 9.14) (https://salilab.org/modeller/). The model obtained was subjected to model validation by using ProSA-web. A structural figure was made with PyMol (ver. 2.4.0, Schrödinger, Inc., New York, NY, USA).

## 3. Results

### 3.1. aaRS Genes in Arabidopsis

We chose the Arabidopsis genome in which to study plant aaRS genes, since *A. thaliana* is the most studied plant among plant research communities, and rich data resources are available. AaRS and related genes were retrieved from the Arabidopsis database by annotation searches and BLAST homology searches. The results are shown in Table 1.

In total, 59 aaRS-related sequences were identified. This is one of the largest sets of aaRS genes found in a single organism. In most organisms, eukaryotic aaRSs can be classified into two sets of genes, a cytosolic set, and an organelle-associated set. In plants, dual targeting of aaRSs is commonly observed, such as to mitochondria and chloroplasts, or to the cytosol and mitochondria [18]. Five aaRSs (arginyl-tRNA synthetase (ArgRS), glutamyl-tRNA synthetase (GluRS), LeuRS, TrpRS, and SerRS) were represented by two genes, while the others were represented by more than two genes. AsnRS possessed the largest number of genes at six. One of the TyrRS (At1g28350) genes was double the expected length, containing two full-length coding sequences fused by a linker, an organization also found in nematodes [19,20]. Surprisingly, 14 fragmented sequences were identified. Most of them display a large contiguous deletion, either from the N- or C-terminus. Some retain only a small portion of the full-length sequence. Eleven aaRSs (glutaminyl-tRNA synthetase (GlnRS), isoleucyl-tRNA synthetase (IleRS), MetRS, valyl-tRNA synthetase (ValRS), alanyl-tRNA synthetase (AlaRS), AsnRS, glycyl-tRNA synthetase (GlyRS), HisRS, lysyl-tRNA synthetase (LysRS), prolyl-tRNA synthetase (ProRS), and threonyl-tRNA synthetase (ThrRS)) were represented among these fragmented sequences.

Genes for the aaRS-related proteins, stand-alone editing domains of AlaRS (AlaXp) and ProRS (ProXp-ala), D-tyrosyl-deacylase (DTD), Asn amidotransferase, three subunits of Gln amidotransferase (GatCAB), and aaRS associated protein (Arc1p) were all identified. In plants, the indirect pathway for Asn- and Gln-tRNA formation operates in the mitochondria [21].

### 3.2. Fragmented Sequences of aaRSs Found in the Arabidopsis Genome

The Arabidopsis genome revealed many fragmented sequences of aaRSs, which lacked minor to major portions of the full-length sequences. In total, 14 such sequences were identified.

Arabidopsis has six AsnRS genes, three of which are cytosolic and contain a WHEP domain insertion, one is organelle specific, and the other two represent only a portion of the full-length sequence (see below Figure 3b). At1g68420 comprises 87 residues, corresponding to motif 2, one of the three conserved signature motifs among class II aaRSs, and a downstream region only. At5g38750 comprises 135 residues, corresponding to a part of the anticodon binding domain (ABD) and motif 1. Motif 1 is located at the dimer interface while motifs 2 and 3 are located at the catalytic site of class II aaRSs. The two fragmented AsnRS sequences were more similar to cytosolic ones, SYNC1 and SYNC3 (see below), than the organellar form.

Arabidopsis has three HisRS genes, one cytosolic, one organellar, and one, At5g03406, containing a sequence corresponding to motif 1 and motif 2, but lacking the C-terminal remainder, including motif 3 (see below Figure 4b).

Arabidopsis has 4 ThrRS genes, two of which represent partial sequences (Figure 1). At1g17960 has a large deletion in part of an editing domain, and lacks the entire motif 1. At1g18130 comprises a C-terminal sequence corresponding to motif 3 and an ABD.

Arabidopsis has two incomplete GlyRS genes (Figure 1). At1g29870 lacks a major portion of the C-terminus, including an ABD, while At3g44740 has a sequence corresponding to motifs 2 and 3, but lacks the motif 1. At1g49930 of AlaRS comprises 133 residues and corresponds to C-terminus of the full-length (Figure 1). Similarly, At5g19720 of GlnRS comprises 170 residues and also corresponds to C-terminus of the full-length (Figure 1). On the other hand, At5g02680 of MetRS comprises 166 residues and corresponds to a middle portion of the full-length, partially overlapping with an endothelial monocyte activating polypeptide II (EMAPII) domain (Figure 1).

One of the three LysRS genes, At3g30805, encodes the C-terminus, containing only motif 3 (Figure 1). Similarly, At5g10880 of ProRS encodes a C-terminal region including only an ABD (Figure 1). At1g27160 from a class I ValRS encodes a C-terminal region corresponding to part of an ABD (Figure 1). Finally, At3g23145 of IleRS encodes 158 residues corresponding only to an editing domain (Figure 1).

To determine whether some of these fragmented sequences are conserved in other plants, we searched for aaRS genes in the rice genome for comparison (Appendix A). A total of 62 genes were identified. Among them, seven gene fragments were found. These were from cysteinyl-tRNA synthetase (CysRS), IleRS, LeuRS, MetRS, TrpRS, TyrRS, and ValRS, all of which are class I. Only IleRS, MetRS, and ValRS were found to be shared with Arabidopsis. However, no relationship was found between the fragmented sequences of the two plants.

### 3.3. Plant-Specific Domains Found in Arabidopsis aaRSs

A summary of the characteristic domains found among Arabidopsis aaRSs is shown in Figure 2. The GST domain (GluRS), WHEP domain (GlyRS), C-terminal extension (SerRS), DNA binding domain (phenylalanyl-tRNA synthetase α-subunit (PheRSα)), TGS domain (a domain commonly found in ThrRS, GTPases, and guanosine polyphosphate hydrolase (SpoT)) (ThrRS), and RNA binding domain (GlnRS) of plant and human aaRSs were found to be related to each other [22]. By contrast, the WHEP domains of HisRS, TrpRS, GluProRS, and MetRS, the GST domains of CysRS, MetRS, and ValRS, the EMAPII domain of TyrRS, and the leucine-zipper motif in arginyl-tRNA synthetase (ArgRS) in human proteins, were not found in Arabidopsis. However, a WHEP domain was present in Arabidopsis AsnRS, and a 400-residue stretch of sequence was present at the N-terminus of HisRS instead of a WHEP domain. In addition, an EMAPII domain was present in MetRS [23]. These features have not been observed in these enzymes’ human counterparts.

### 3.4. Plant-Specific WHEP Domain Insertion in AsnRS

Two aaRSs, AsnRS, and HisRS, contain domains not present in human enzymes. AsnRS is a class II aaRS with a WHEP domain insertion in its catalytic domain between motifs 1 and 2 (Figure 3a). This insertion has been found in all three cytosolic forms, designated as SYNC1 (At5g56680), SYNC2 (At3g07420), and SYNC3 (At1g70980), but is absent from the organellar form designated as SYNO (At4g17300) (Figure 3b) [24]. In fact, this WHEP insertion has only been seen in plants among all kingdoms of life. This is the first case in which a WHEP domain is found in AsnRS, and is the first case in which the WHEP domain is inserted within a catalytic domain. Previously, WHEP domains had only been found either at the N- or C-termini of full-length aaRSs, or in a linker region between GluRS and ProRS in a human bifunctional GluProRS [22]. GlyRS is the only other plant aaRS that contains a WHEP domain. Phylogenetic analysis of WHEP domains showed that plant sequences cluster together, and are distantly related to human sequences as supported by high bootstrap values (Appendix A). A closer look inside sequences from land plants revealed that two clades were formed, one including SYNC1 and SYNC3, and the other includes SYNC2 from Arabidopsis (Appendix A). This is in accordance with an existence of two slightly different full-length cytosolic AsnRS sequences supported by high bootstrap values (Appendix A). Both exhibited moderate sequence identity (44% between SYNC3 and SYNC2), while high sequence identity (83% between SYNC1 and SYNC3) was observed within the same clade. Importantly, this WHEP insertion in AsnRS is highly conserved among plants and, as is the case in Arabidopsis, only the cytosolic forms contain this insertion (Figure 3c). The sequence identity of WHEP domain among plants ranged from 30% to 40%, while the sequence length was conserved to within 60 residues. Therefore, these WHEP domains have conferred an evolutionary advantage during plant evolution. Molecular modeling of cytosolic AsnRS (SYNC3) suggests that the WHEP domain can be modeled next to the catalytic domain (Figure 3d).

### 3.5. Plant-Specific Long N-Terminal Extension in HisRS

Arabidopsis HisRS (At3g02760) was found to contain a long N-terminal extension sequence of ca. 400 residues not seen in the human homolog (Figure 4a). This large extension occurs only in the cytosolic form, not in the organellar form (At3g46100) (Figure 4b). Moreover, it has only been observed in plants and not in other organisms. The extra sequence is predicted to be a histidine ammonia lyase (HAL) by InterPro analysis. The sequence exhibited very weak homology (15–20%) with some parts of HAL. HAL belongs to a lyase family that is comprised of HAL, phenylalanine ammonia lyase (PAL), and tyrosine ammonia lyase (TAL) [25]. These enzymes catalyze the deamination of histidine, phenylalanine, and tyrosine, yielding urocanic acid, cinnamic acid, and *p*-coumaric acid, respectively. The reaction involves the cofactor 3,5-dihydro-5-methylidene-4*H*-imidazol-4-one (MIO), which is formed from conserved Ala, Ser, and Gly residues [26]. HAL catalyzes an initial step in the catabolic pathway that converts histidine to glutamine. Plants do not possess HAL. Its homolog PAL plays a very important role in phenylpropanoid biosynthesis, including flavonoids and lignins. High conservation of the HAL-like domain observed in HisRS in Arabidopsis is seen among plant cytosolic HisRSs, mostly in monocots and dicots but also in some bryophytes and algae (Figure 4c). The sequence identity of the catalytic domain (miniHRS) is approximately 70–80%, while that of the HAL-like domain is in the 30–40% range. The low sequence conservation in the HAL-like domain and the fact that the HAL-like domains seen in plant HisRSs do not contain the MIO forming Ala-Ser-Gly residues suggest that this domain most likely does not possess HAL catalytic activity, but may play a structural role, or support protein-protein interaction with as yet unknown protein partners. Phylogenetic analysis of both HAL-like domain and the miniHRS part showed that both phylogenetic trees were similar to each other (Appendix A). This indicated that both HAL-like domain and the miniHRS part took the same evolutionary path suggesting that both were connected to each other from the very beginning.

## 4. Discussion

Arabidopsis contains 59 aaRS genes. This is one of the largest numbers of these sequences found in a single organism. Plants usually have more genes than vertebrates or mammals. This is partly due to the presence of larger gene families than in other eukaryotes. The large number of aaRS genes in Arabidopsis is mainly due to the presence of multiple genes for each aaRS. However, many of these are fragmented sequences lacking minor to major portions of the full-length sequence.

Our survey of Arabidopsis aaRSs identified characteristic domains found only in plants. A unique WHEP insertion was seen in the cytosolic form of AsnRS. High conservation among plants indicates that it has an indispensable canonical or non-canonical role in AsnRS function in the cytosol. Cytosolic AsnRS is predicted to have experienced a unique evolutionary history. Sequence comparison of plant cytosolic AsnRS shows a closer relationship with mitochondrial AsnRS, suggesting organellar origin [24]. During and after the symbiosis of cyanobacteria ancestral to the chloroplast, organellar AsnRS may have replaced the now extant cytosolic form present at that time, an event called “cytosolic capture”. Acquisition of a WHEP domain presumably took place concurrent with this event, which may have conferred an evolutionary advantage, allowing replacement of the former cytosolic AsnRS. The role of this WHEP domain is unknown. Since other WHEP domains are mainly used for interaction with RNA or protein, this cytosolic AsnRS may interact with RNA or protein(s) to exert its presumed non-canonical functions.

We also found a large domain showing weak homology with HAL attached to the N-terminus of the plant cytosolic HisRS. This HAL-like domain is highly conserved among cytosolic HisRSs in land plants, indicating its important role in HisRS function. Because it lacks the residues to form the cofactor MIO, this HAL-like domain is probably not catalytic, and may have another role, such as maintaining the structural integrity of HisRS, or interacting with other protein partners. In fact, sequence comparison across plants shows that the HAL-like domain has lower sequence conservation than the miniHRS catalytic domain. HAL is absent from plants, but belongs to the same enzyme family as PAL, an abundant and prevalent enzyme in plants. These enzymes form tetramers. Therefore, it is possible that this HAL-like domain may interact with PAL. Since PAL plays an important role in plants in the biosynthesis of flavonoids, anthocyanins, and phenylpropanoids, including lignins, HisRS may play a regulatory role in production of these plant metabolites.

Our finding that two aaRSs have unique domains found only in plants suggests that these domains have non-canonical functions in plants. Many non-canonical functions of mammalian proteins are exhibited through species-specific domains [22]. The high conservation of these domains among plants strongly implies their indispensable role in aaRS functions. Future studies will be required to uncover novel functions related to these domains in plant aaRSs.

In addition to aaRSs possessing plant-specific domains, we identified a number of fragmented aaRS sequences in the Arabidopsis genome. These genes possess only a portion of a full-length sequence, indicating that they are catalytically null fragments. Whether such genes have physiological functions or represent pseudogenes must await future studies. Since these fragments were not conserved among plants, at least between Arabidopsis and rice, the occurrence of such genes appears to be species-specific within plants. The large number of SVs identified in humans, many of which lack catalytic domains, implies that aaRS fragments might have physiological functions themselves [6]. Thus, the fragmented sequences found in the plant genome may also possess functions. SVs of plant aaRSs have not been studied so far, and we do not know the extent to which SVs exist in plant aaRSs. Some sequences corresponding to a particular SV may have been coded separately in the genome for important functions they carry out in plants. These gene fragment sequences are worthy of further investigation in the future.

The sequence of the IleRS *At3g23145* encodes only the editing domain. Stand-alone editing domains are well known in AlaRS, known as AlaXp [27], in ProRS known as ProXp-ala [28] and ProXp-ST [29], and in ThrRS [30]. These proteins support the editing of misacylated tRNAs in trans to maintain fidelity in protein synthesis. Such a stand-alone editing domain has not been found in any IleRS; thus, At3g23145 may be the first example. It is interesting to note that a connective polypeptide 1 (CP1) regions of IleRS and ValRS that correspond to the editing domain were able to be expressed alone and exhibited editing activity in vitro in trans [31]. However, the sequence of At3g23145 lacks a portion of the domain, and critical residues conferring the ability to hydrolyze misacylated tRNAs [32]. Therefore, At3g23145 may not possess editing activity and may play a different role.

In addition, *At1g27160* of ValRS encodes only a C-terminal ABD. Thus, it can be regarded as a stand-alone ABD. Whether such a protein plays a role in the aminoacylation of tRNA^Val^ or has a completely different role is of interest

In HisRS, a C-terminal truncated *hisZ* gene is known [33,34]. HisZ participates in histidine biosynthesis, which codes for a regulatory subunit of adenosine 5’-triphosphate phosphoribosyltransferase (ATPPRT) and forms a complex with HisG to yield the holoenzyme. Arabidopsis At5g03406 seems to have a similar structure to that of HisZ, but lacks the entire coding region for the class II motif 3.

## 5. Conclusions

A survey of the Arabidopsis genome revealed 59 aaRS sequences. Among them, AsnRS and HisRS were found to contain plant-specific domains, suggesting that they have plant-specific non-canonical roles. A number of gene fragments encoding only portions of the full-length enzymes were found. These genes may code for catalytically null fragments of aaRSs that may fulfill other plant specific roles. The identified genes should serve as top candidates to explore non-canonical functions in plants, which are largely unknown. In accordance with higher eukaryotes’ aaRSs possessing more non-canonical functions, plants are expected to possess some such functions, some of which are likely to be unique to plants. A completely different lifestyle in plants compared to vertebrates may have driven the evolution of novel non-canonical functions not seen before, and may further illuminate the potential of aaRSs to carry out different tasks in different biological backgrounds. By exploring aaRSs’ non-canonical functions, we may uncover novel plant functions that could not be discovered with conventional approaches.

## Figures and Tables

**Figure 1 genes-11-01056-f001:**
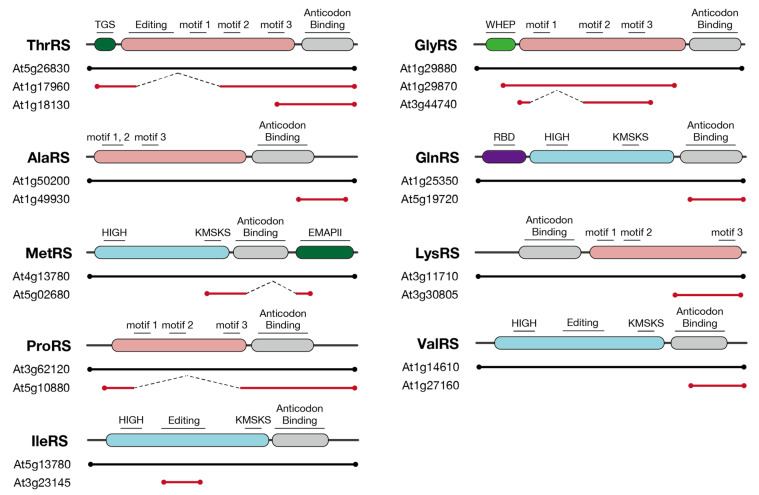
Fragmented sequences of aaRSs found in Arabidopsis genome. Each fragmented sequence (red bar) was compared with a full-length cytoplasmic sequence (black bar) to show the relative position among the full-length sequence.

**Figure 2 genes-11-01056-f002:**
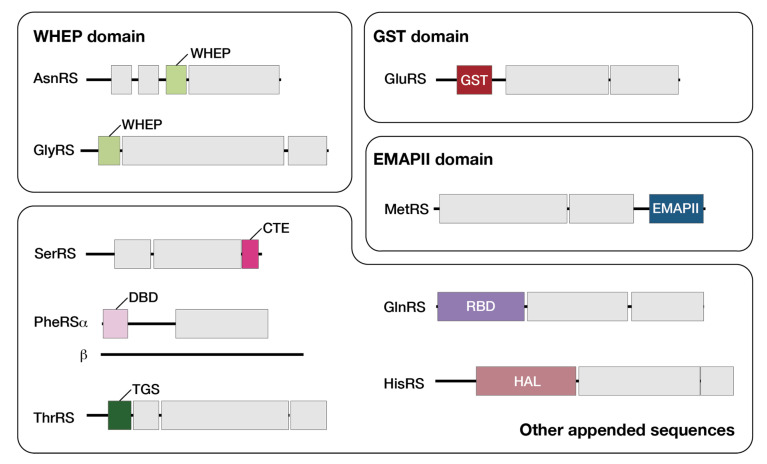
Characteristic domains found in Arabidopsis aaRSs. Abbreviations: CTE, the C-terminal extension; DBD, DNA binding domain; GST, glutathione *S*-transferase; RBD, RNA binding domain; TGS, the TGS domain named after ThrRS, GTPases, and SpoT; HAL, histidine ammonia lyase-like domain; EMAPII, endothelial monocyte activating polypeptide II domain.

**Figure 3 genes-11-01056-f003:**
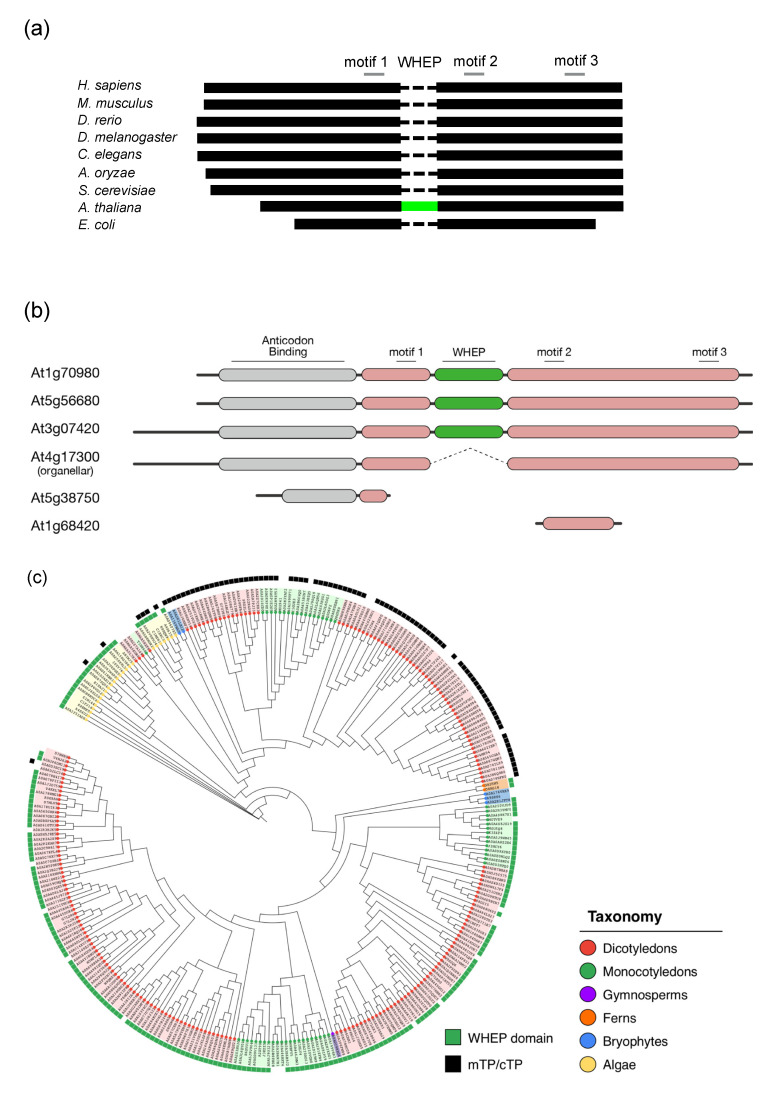
AsnRS from plants. (**a**) Schematic representation of the full-length AsnRS from various organisms. WHEP domain (green) is only seen inserted within a catalytic domain in plants. Three conserved motifs among class II aaRSs are shown. Motif 1 is located at the dimer interface while motifs 2 and 3 are located at the catalytic site. (**b**) Schematic representation of all the six AsnRS sequences in Arabidopsis. (**c**) Phylogenetic tree of AsnRS from plants. Green-colored boxes indicate those containing a WHEP domain. Black boxes indicate those containing mitochondrial (mTP) or chloroplast transit peptide (cTP) sequences, and hence are organellar forms. Sequences from different taxonomy are colored differently. (**d**) Molecluar modeling of Arabidopsis AsnRS (SYNC3). Structures of AsnRS from *P. horikoshii* (1X54) and a WHEP domain from human TrpRS (1R6T) were used as templates to build SYNC3 structure. Structure of AspRS bound with tRNA (1EFW) was used as a template for modeling bound tRNA. Each domain is colored differently.

**Figure 4 genes-11-01056-f004:**
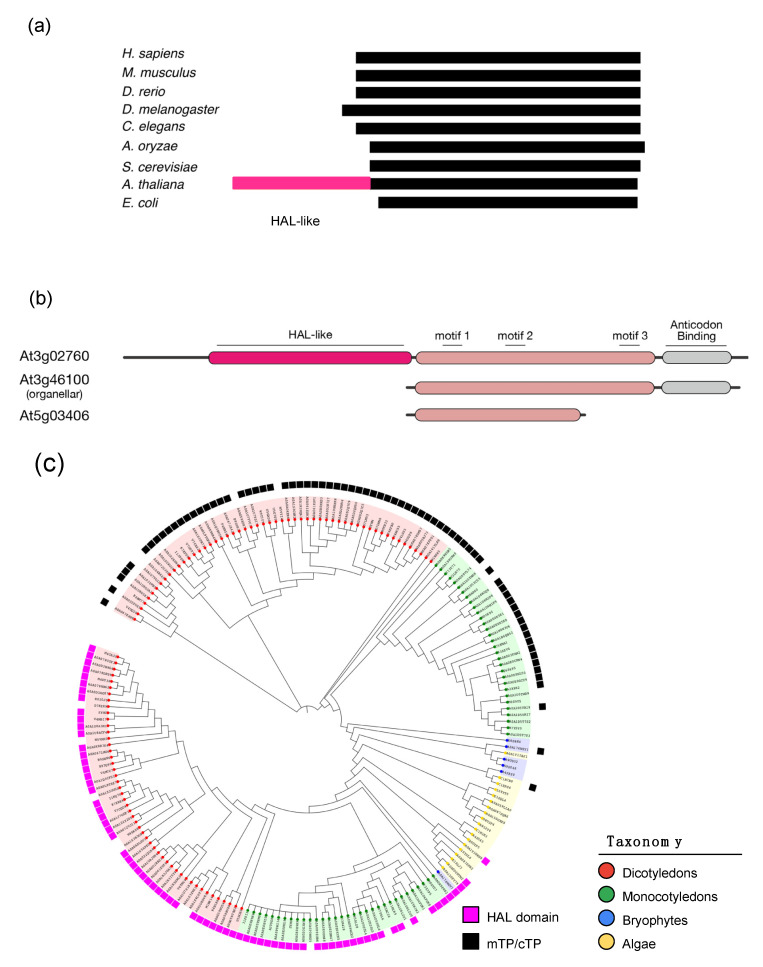
HisRS from plants. (**a**) Schematic representation of the full-length from various organisms. HAL-like domain (red) is only seen in plants. (**b**) Schematic representation of all three HisRS sequences in Arabidopsis. (**c**) Phylogenetic tree of HisRS from plants. Magenta-colored boxes indicate those containing a HAL-like domain in addition to the full-length catalytic domain. Black boxes indicate those containing mitochondrial (mTP) or chloroplast transit peptide (cTP) sequences, and hence are organellar forms. Sequences from different taxonomy are colored differently.

**Table 1 genes-11-01056-t001:** Arabidopsis aaRSs identified in the database. Characteristic primary structures are noted.

Class I	AGI No.	Localization Prediction	Primary Structure	Class II	AGI No.	Localization Prediction	Primary Structure
ArgRS	At4g26300	cytosol, cp		AlaRS	At1g50200	cytosol, mt	
	At1g66530	cytosol, mt			At5g22800	mt, cp	
CysRS	At5g38830	cytosol			At1g49930		N truncated
	At3g56300	cytosol		AsnRS	At5g56680	cytosol	
	At2g31170	mt, cp			At4g17300	mt, cp	
GlnRS	At1g25350	cytosol			At1g70980	cytosol	
	At5g19720		N truncated		At3g07420	cytosol	
GluRS	At5g26710	cytosol			At1g68420		N, C truncated
	At5g64050	mt, cp			At5g38750		N, C truncated
IleRS	At5g49030	mt, cp		AspRS	At4g33760	mt, cp	
	At4g10320	cytosol			At4g31180	cytosol	
	At3g23145		N, C truncated		At4g26870	cytosol	
LeuRS	At4g04350	cp		GlyRS	At3g48110	mt, cp	
	At1g09620	cytosol, mt			At1g29880	cytosol, mt	
MetRS	At4g13780	cytosol			At3g44740		N, C truncated
	At3g55400	mt, cp			At1g29870		C truncated
	At5g02680		N, C truncated	HisRS	At3g02760	cytosol	
TrpRS	At3g04600	cytosol			At3g46100	mt, cp	
	At2g25840	mt, cp			At5g03406		C truncated
TyrRS	At3g02660	mt, cp		LysRS	At3g11710	cytosol	
	At2g33840	cytosol			At3g13490	mt, cp	
	At1g28350	cytosol	pseudo-dimer		At3g30805		N truncated
ValRS	At5g16715	mt, cp		PheRS	At4g39280	cytosol	
	At1g14610	cytosol, mt			At1g72550	cytosol	
	At1g27160		N truncated		At3g58140	mt, cp	
				ProRS	At5g52520	mt, cp	
					At3g62120	cytosol	
					At5g10880		N truncated
				SerRS	At5g27470	cytosol	
					At1g11870	mt, cp	
				ThrRS	At5g26830	cytosol, mt	
					At2g04842	mt, cp	
					At1g18130		N truncated
					At1g17960		motif1 missing

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
