# Peer review of "Plant-Specific Domains and Fragmented Sequences Imply Non-Canonical Functions in Plant Aminoacyl-tRNA Synthetases"

_genes, 2020, doi:10.3390/genes11091056_

Round 1

Reviewer 1 Report

Summary:

This manuscript describes a largely bioinformatics analysis of plant aminoacyl-tRNA synthetases (aaRSs) and synthetase-like domains, an understudied area. Numerous aaRSs (mostly in vertebrates) have been discovered to possess a range of non-canonical functions but few studies have been carried out in plant aaRS. The authors identified a total of 59 aaRS genes in Arabidopsis, one of the largest sets of aaRSs in a single organism. They provide a summary and overview of these genes and focus on several aspects in somewhat more detail:

  • AsnRS, which has a WHEP domain inserted into the catalytic site in the cytosolic AsnRS genes. A homology model of this aaRS was generated.
  • HisRS, which has a domain with homology to histidine ammonia lyase (HAL) at the N-terminus of cytosolic plant HisRS.
  • aaRS fragments. Attempts to obtain soluble aaRS fragments upon recombinant protein expression in coli are also reported for a subset of these plant fragments.

Overall, this is a worthwhile study and lots of interesting information is included in the bioinformatics analyses, Tables and some of the Figures. Some of the conclusions are speculative and should be revised and some of the figures could be improved. The recombinant protein expression tests are not well developed, do not lead to any biological insights, and should be deleted. Detailed comments and suggestions are given below.

Major comments:

  1. The following sentences in the abstract should be revised or deleted:

Line 28: These genes encode catalytically null aaRS fragments. Change to: These genes encode 29 aaRS fragments that lack key active site sequences and are likely catalytically null.

Delete: Soluble proteins have been obtained for some of 30  the fragment genes by heterologous expression in E. coli, suggesting that when expressed in plants these genes may have novel functions. Just because they are soluble in E. coli, doesn’t mean they have novel functions. See also below on strong suggestion to delete these experiments as they do not allow any conclusions to be made. Many plant proteins are difficult to express in E. coli and that doesn’t tell us anything about their potential function in plants. 

  1. Many abbreviations are used that are not defined, including HisRS and all aaRSs first time mentioned, WHEP domain (what is this? Most non-synthetase experts will have never heard of this and it needs a very clear explanation the first time it appears), GST, PARP-1, TGS, etc. The list is very long. Also, what are functions of motifs 1, 2 and 3?
  2. Methods section: homology modeling methods should be described in more detail in this section. I am recommending the E. coli expression section be deleted as this data does not provide any significant biological insights.
  3. Line 62: The authors may want to list additional known plant non-canonical functions:

Zuo DY, Yi SY, Liu RJ, Qu B, Huang T, He WJ, Li C, Li HP & Liao YC (2016) Deoxynivalenol-activated methionyl-tRNA synthetase gene from wheat encodes a nuclear localized protein and protects plants against Fusarium pathogens and mycotoxins. Phytopathology 106, 614–623.

=> mycotoxin deoxynivalenol activates the expression of wheat MetRS which can lead plant defense and detoxification conferring resistance to mycotoxins and mycotoxin-producing fungal pathogens

Yang X, Li G, Tian Y, Song Y, Liang W & Zhang D (2018) A rice glutamyl-tRNA synthetase modulates early anther cell division and patterning. Plant Physiol 177, 728–744.

=> Rice GluRS regulate male organ development by affecting metabolic homeostasis and redox status.

  1. Line 123: Can the authors check if duel targeting to the “chloroplast and cytosol” really occurs?
  2. 6. In lines 133-137, are these aaRS-related proteins meant to be exhaustive? Are there any homologs of ProRS or ThrRS editing domains? AlaXp is mentioned but what about ProXp-ala or other ProX homologs?
  3. Line 166: Phylogenetic analysis of WHEP domains showed that plant sequences cluster together, and are distantly related to human sequences. This analysis should be shown in a figure.
  4. Line 180. the WHEP domain protrudes from the catalytic domain and may interact with acceptor stem of tRNA. This statement is so speculative it should be deleted.
  5. Figure 2c is never discussed or cited in the text.
  6. Line 230: Any explanation for why AsnRS needs 3 cytosolic genes for cytosolic alone? Are the AsnRS fragments more similar to cytosolic or organeller?
  7. Line 216-291: states while that of the HAL-like domain is in the 30% –40% range. The low sequence conservation in the HAL-like domain and the fact that the HAL-like domains seen in plant HisRSs do not contain the MIO forming Ala-Ser-Gly residues suggest that the actual sequence might not be important. This is actually rather high identity (or at least moderate) and doesn’t indicate the sequence is not important. Remove or rephrase.
  8. Line 226: Section 3.4 should be moved earlier to improve the flow of the manuscript. I suggest this section and accompanying figures move after Table 1. Also, if the short 1-sentence paragraphs could be combined and revised to flow better, that would help.
  9. Section 3.5 doesn’t provide any useful information that adds to this manuscript and the conclusions drawn from this data do not make sense. This should be removed from the current study.
  10. Line 343 states: The sequence of the IleRS At3g23145 encodes only the editing domain. A stand-alone editing domain is well known in AlaRS, known as AlaXp [25]. This protein supports the editing of misacylated tRNAAla in trans to maintain fidelity in protein synthesis. There are other known editing domains derived from class 2 aaRSs (ProRS and ThrRS) that could also be mentioned.

It is indeed interesting that there may be a free-standing class I CP1-like domain in plants. A long time ago, the Schimmel lab showed that CP1 domains can be expressed alone and display editing activity in vitro and these papers could also be mentioned/referenced.

  1. The quality of some of the figures, including most of the supplementary figures is poor. The figure legends need to be more detailed especially in the case of the supplementary figures, and the legends should be with the figures.

Specific comments on the figures:

Figure 2a: revise the green words WHEP domain to match other labels like motif 1 in this figure. I don’t think motif 1,2,3 were ever defined.

Figure 3a: revise the red words HAL-like domain to match other labels in the figure.

Figure 4 legend: mention that only cytoplasmic sequences are shown. “full-length cytoplasmic sequence (black bar)”

Fig S2 and S3: what do the sizes of the blue circles indicate exactly?

Figure S4 could be improved visually because the branches and labels are a bit awkwardly spaced. I also think enhancing that figure is necessary to provide convincing evidence for the claim that the HAL-like domain and the miniHRS “took part in the same evolutionary path”.

Table S2: could the localization prediction also be included?

Minor comments:

“Faint” homology used in several places should be changed to weak or low homology

“Built-in” co-factor should just be co-factor

Many grammatical errors (for example, lines 91 and 108)—suggest careful proofreading by an English expert.

line

19        V = Very?

41        Omit comma

42        exerted;  consider “exhibited” instead?

147      mention these are human proteins

181:     2c is cited here and it should be 2d

201      This should be a reference to Figure 3b

214      This should be a reference to Figure 3c

255      missing “(Fig.4)”

Author Response

 >First of all, I sincerely appreciate for taking your precious time to evaluate our manuscript and also for a lot of useful comments to improve our manuscript. We have now revised our manuscript based on all these comments. Largely, we deleted all the E. coli expression experiments, as we agree with the referee that these data do not lead to any major conclusions. Also, section 3.6 was moved to 3.2 to make it a better flow in the manuscript. Followings are detailed point-by-point responses.

Summary:

Overall, this is a worthwhile study and lots of interesting information is included in the bioinformatics analyses, Tables and some of the Figures. Some of the conclusions are speculative and should be revised and some of the figures could be improved. The recombinant protein expression tests are not well developed, do not lead to any biological insights, and should be deleted. Detailed comments and suggestions are given below.

Major comments:

  1. The following sentences in the abstract should be revised or deleted:

Line 28: These genes encode catalytically null aaRS fragments. Change to: These genes encode 29 aaRS fragments that lack key active site sequences and are likely catalytically null.

>Thank you very much for the suggestions. The sentence has been changed (Line 29).

Delete: Soluble proteins have been obtained for some of 30 the fragment genes by heterologous expression in E. coli, suggesting that when expressed in plants these genes may have novel functions. Just because they are soluble in E. coli, doesn’t mean they have novel functions. See also below on strong suggestion to delete these experiments as they do not allow any conclusions to be made. Many plant proteins are difficult to express in E. coli and that doesn’t tell us anything about their potential function in plants.

>I agree that our E. coli expression studies are rather supplement and do not lead to any major conclusions, and thus, will delete this section and related sentences as well.

  1. Many abbreviations are used that are not defined, including HisRS and all aaRSs first time mentioned, WHEP domain (what is this? Most non-synthetase experts will have never heard of this and it needs a very clear explanation the first time it appears), GST, PARP-1, TGS, etc. The list is very long. Also, what are functions of motifs 1, 2 and 3?

>All the abbreviations are now defined at their first appearance. I apologize for not following the format. Also, motifs 1, 2, and 3 of class II aaRSs are now defined at their first appearance in 3.2 (Line 156 and 159).

  1. Methods section: homology modeling methods should be described in more detail in this section. I am recommending the E. coli expression section be deleted as this data does not provide any significant biological insights.

>Homology modeling methods are now described in more detail (Line 103~110) and all the E. coli expression experiments are deleted.

  1. Line 62: The authors may want to list additional known plant non-canonical functions:

Zuo DY, Yi SY, Liu RJ, Qu B, Huang T, He WJ, Li C, Li HP & Liao YC (2016) Deoxynivalenol-activated methionyl-tRNA synthetase gene from wheat encodes a nuclear localized protein and protects plants against Fusarium pathogens and mycotoxins. Phytopathology 106, 614–623.

=> mycotoxin deoxynivalenol activates the expression of wheat MetRS which can lead plant defense and detoxification conferring resistance to mycotoxins and mycotoxin-producing fungal pathogens

Yang X, Li G, Tian Y, Song Y, Liang W & Zhang D (2018) A rice glutamyl-tRNA synthetase modulates early anther cell division and patterning. Plant Physiol 177, 728–744.

=> Rice GluRS regulate male organ development by affecting metabolic homeostasis and redox status.

>Thank you very much for the suggestions. The two mentioned references are now included as ref 11 and 17 and appropriate text is added (Line 73~75).

  1. Line 123: Can the authors check if duel targeting to the “chloroplast and cytosol” really occurs?

>I am afraid we are not an expert on targeting and only have to rely on literature data. But indeed, dual targeting is an interesting observation and its mechanism is worth investigations. Careful inspection of the literature in reference 18 indicated that 15 aaRSs were shared between mitochondria and chloroplast, 5 were shared between cytosol and mitochondria, and 1 was between cytosol and chloroplast. Therefore, the sentence was changed to "mitochondria and cytosol" (Line 128).

  1. In lines 133-137, are these aaRS-related proteins meant to be exhaustive? Are there any homologs of ProRS or ThrRS editing domains? AlaXp is mentioned but what about ProXp-ala or other ProX homologs?

>I am terribly sorry not to mention ProXps. ProXp-ala is now added in the list (Line 144). We could not find any ThrRS editing domain homolog in Arabidopsis.

  1. Line 166: Phylogenetic analysis of WHEP domains showed that plant sequences cluster together, and are distantly related to human sequences. This analysis should be shown in a figure.

>This is now shown in Supplementary figures Fig. S1 and S2.

  1. Line 180. the WHEP domain protrudes from the catalytic domain and may interact with acceptor stem of tRNA. This statement is so speculative it should be deleted.

>The statement is now deleted (Line 246).

  1. Figure 2c is never discussed or cited in the text.

>I am sorry for the mistake. All the figures are now mentioned in the text.

  1. Line 230: Any explanation for why AsnRS needs 3 cytosolic genes for cytosolic alone? Are the AsnRS fragments more similar to cytosolic or organeller?

>This is a very intriguing question, which obviously does not have an answer. Phylogenetic tree clearly indicate that the three are separated into two groups (SYNC1/3 and SYNC2) (Fig. S4, S5). The two have different length and sequence identity is rather low. I believe that there are some functional differences between the two groups. On the other hand, the two AsnRS fragments were more similar to cytosolic form and this is now indicated in the text (Line 160~161).

  1. Line 216-291: states while that of the HAL-like domain is in the 30% –40% range. The low sequence conservation in the HAL-like domain and the fact that the HAL-like domains seen in plant HisRSs do not contain the MIO forming Ala-Ser-Gly residues suggest that the actual sequence might not be important. This is actually rather high identity (or at least moderate) and doesn’t indicate the sequence is not important. Remove or rephrase.

>Thank you very much for the suggestion. The "sequence is not important" statement is now removed both from this section (Line 288) and from discussion section (Line 334).

  1. Line 226: Section 3.4 should be moved earlier to improve the flow of the manuscript. I suggest this section and accompanying figures move after Table 1. Also, if the short 1-sentence paragraphs could be combined and revised to flow better, that would help.

>This section 3.4 is now moved to 3.2. Accordingly, Fig. 4 is now changed to Fig. 1. The short 1-sentence paragraphs are now combined.

  1. Section 3.5 doesn’t provide any useful information that adds to this manuscript and the conclusions drawn from this data do not make sense. This should be removed from the current study.

>I agree with the suggestion and this section is now removed.

  1. Line 343 states: The sequence of the IleRS At3g23145 encodes only the editing domain. A stand-alone editing domain is well known in AlaRS, known as AlaXp [25]. This protein supports the editing of misacylated tRNAAla in trans to maintain fidelity in protein synthesis. There are other known editing domains derived from class 2 aaRSs (ProRS and ThrRS) that could also be mentioned.

It is indeed interesting that there may be a free-standing class I CP1-like domain in plants. A long time ago, the Schimmel lab showed that CP1 domains can be expressed alone and display editing activity in vitro and these papers could also be mentioned/referenced.

> I am terribly sorry not to mention ProXps. Stand-alone ProXps and ThrRS editing domain are now added in the text and their references (# 28~30) as well (Line 364). Also, thank you very much for suggesting the literature. I wasn't aware of the work, which I should have. This is now mentioned in the text and reference added (#31) (Line 367~370).

  1. The quality of some of the figures, including most of the supplementary figures is poor. The figure legends need to be more detailed especially in the case of the supplementary figures, and the legends should be with the figures.

>I apologize for the figure quality. Most of the figures are now revised and the figure legends for supplementary figures are extensively revised. Also the legends for supplementary figures are now included in the figures.

Specific comments on the figures:

Figure 2a: revise the green words WHEP domain to match other labels like motif 1 in this figure. I don’t think motif 1,2,3 were ever defined.

>The words WHEP domain is now revised. Motif 1, 2, 3 of class II aaRSs is now defined both in the text and in the legend.

Figure 3a: revise the red words HAL-like domain to match other labels in the figure.

>This is now revised.

Figure 4 legend: mention that only cytoplasmic sequences are shown. “full-length cytoplasmic sequence (black bar)”

>This is now revised (Line 208).

Fig S2 and S3: what do the sizes of the blue circles indicate exactly?

>The blue circles are now omitted and instead, bootstrap values are indicated in percentage (Fig. S1, S2, S4, S5).

Figure S4 could be improved visually because the branches and labels are a bit awkwardly spaced. I also think enhancing that figure is necessary to provide convincing evidence for the claim that the HAL-like domain and the miniHRS “took part in the same evolutionary path”.

>Thank you very much for the suggestions. The phylogenetic trees are now revised and I believe will be more convincing (Fig. S6, S7).

Table S2: could the localization prediction also be included?

>This is now included in the table (Table S2).

Minor comments:

“Faint” homology used in several places should be changed to weak or low homology

>This is now revised (Line 272 and 327).

“Built-in” co-factor should just be co-factor

>This is now revised (Line 277 and 330).

Many grammatical errors (for example, lines 91 and 108)—suggest careful proofreading by an English expert.

line

19       V = Very?

41       Omit comma

42       exerted; consider “exhibited” instead?

147     mention these are human proteins

181:     2c is cited here and it should be 2d

201     This should be a reference to Figure 3b

214     This should be a reference to Figure 3c

255     missing “(Fig.4)”

>I apologize for these careless mistakes. All the above are now revised.

Reviewer 2 Report

This is an excellent paper that presents an integrated computational/experimental analysis of (potentially) non-canonical functions in plant aaRSs. It complements well the existing body of similar work in other species/domains of life. It should be of significant value to researchers interested in aarSs, and to anyone interested in genetic code and translation machinery evolution, in general.

This reviewer is not a lab scientist, so I cannot comment on experimental aspects of the study (such as section 2.2). 

This said, I have major concerns regarding the computational methodology involved, namely the phylogenetic analysis:

First, the authors' choice of the phylogenetic reconstruction method(s), model(s) and corresponding software is not explained or motivated in any way, shape, or form. The authors just list the acronyms. At the very least, the authors should explain (i) why these methods and models (or, rather, automated model-fitting approaches) are appropriate for this type of data, (ii) why they are robust to explicit and implicit method/model assumption violations, and (iii) how do they compare to the choice of methods/models in the literature (we're talking about earlier, comparable, phylogenetic studies of aaRSes). 

Second, the "blue circles" indicating bootstrap values in suppl. figures are really not informative at all. The authors should just present the actual bootstrap values, and discuss the significance (or insignificance) of the lower ones (e.g., < 75%) in the context of their conclusions.

Third, given the dimensionality of the data, 100 bootstrap replicates is simply not enough. I'd suggest at least 1,000 or, preferably, 10,000. Again, given the dimensionality of the data, this shouldn't be too computationally demanding.

Fourth, how robust are the authors' results/conclusions to the variations in the phylogenetic analysis framework? What if the authors try (slightly) different methods/models/software? Will the results stands? Frankly, I can't even harbor a ballpark guess at this time, because I don't have the actual bootstrap values in front of me, just the (very few) blue circles. 

My intuition suggests that the results will be sufficiently robust... but it's very difficult to make a substantiated conclusion based on the phylogenetic analysis details currently presented by the authors. 

Other than that, the study is convincing, and the manuscript is well put together. The enthusiasm, however, is dampened by the anemic phylogenetic analysis and its presentation.

Author Response

>I sincerely appreciate for taking your precious time to evaluate our manuscript and also for a lot of useful comments to improve our manuscript. We have now revised our manuscript based on all these comments.

Comments and Suggestions for Authors

This is an excellent paper that presents an integrated computational/experimental analysis of (potentially) non-canonical functions in plant aaRSs. It complements well the existing body of similar work in other species/domains of life. It should be of significant value to researchers interested in aarSs, and to anyone interested in genetic code and translation machinery evolution, in general.

This reviewer is not a lab scientist, so I cannot comment on experimental aspects of the study (such as section 2.2).

This said, I have major concerns regarding the computational methodology involved, namely the phylogenetic analysis:

First, the authors' choice of the phylogenetic reconstruction method(s), model(s) and corresponding software is not explained or motivated in any way, shape, or form. The authors just list the acronyms. At the very least, the authors should explain (i) why these methods and models (or, rather, automated model-fitting approaches) are appropriate for this type of data, (ii) why they are robust to explicit and implicit method/model assumption violations, and (iii) how do they compare to the choice of methods/models in the literature (we're talking about earlier, comparable, phylogenetic studies of aaRSes).

>Thank you very much for the comments. In order to answer points (i) and (ii), we included a Supplementary Table S1 showing the choice and validation of models used for each phylogenetic analysis (7 in total) from ModelTest-NG. For point (iii), this is actually very difficult since most of the papers in the field do not mention in detail the choice of methods/models in their phylogenetic analysis and is impossible to make comparison (for example, Peeters N.M. et al., J. Mol. Evol. 50, 413-423 (2000), Novoa E.M. et al, J. Biol. Chem. 290, 10495-10503 (2015)). Here, we do not intend to invent a new method nor characterize our methods from others in our phylogenetic analysis, but rather to follow conventional approaches. We believe our methods are relatively standard and not quite distinguished from others.

Second, the "blue circles" indicating bootstrap values in suppl. figures are really not informative at all. The authors should just present the actual bootstrap values, and discuss the significance (or insignificance) of the lower ones (e.g., < 75%) in the context of their conclusions.

>I apologize for using such obscure blue circles. The figures are now revised and bootstrap values are indicated in percentage (Fig. S1, S2, S4, S5, S6, S7). The significance and insignificance of the branches are discussed in the legends as well as some in the text.

Third, given the dimensionality of the data, 100 bootstrap replicates is simply not enough. I'd suggest at least 1,000 or, preferably, 10,000. Again, given the dimensionality of the data, this shouldn't be too computationally demanding.

>The analysis is now revised and 1,000 bootstrap replicates are used. In the figure, percentage is indicated.

Fourth, how robust are the authors' results/conclusions to the variations in the phylogenetic analysis framework? What if the authors try (slightly) different methods/models/software? Will the results stands? Frankly, I can't even harbor a ballpark guess at this time, because I don't have the actual bootstrap values in front of me, just the (very few) blue circles.

>To answer this point, we analyzed 4 of our phylogenetic analysis using Neighbor-Joining method. The resulting phylogenetic trees are now shown in Supplementary figures (Fig. S2, S5, S7) for comparison with maximum likelihood methods (Fig. S1, S4, S6). The results are basically the same with the trees obtained from the original maximum likelihood method.

Round 2

Reviewer 2 Report

The manuscript is significantly improved. Now that the 1000 bootstrap replicates and two types of methods (NJ and ML) are present, there is high confidence in the robustness of the results.